# Biomarkers for Managing Neurodegenerative Diseases

**DOI:** 10.3390/biom14040398

**Published:** 2024-03-26

**Authors:** Lara Cheslow, Adam E. Snook, Scott A. Waldman

**Affiliations:** 1Department of Pharmacology, Physiology and Cancer Biology, Thomas Jefferson University, Philadelphia, PA 19107, USA; lara.cheslow@students.jefferson.edu (L.C.); adam.snook@jefferson.edu (A.E.S.); 2Department of Neurosciences, Thomas Jefferson University, Philadelphia, PA 19107, USA; 3Department of Microbiology and Immunology, Thomas Jefferson University, Philadelphia, PA 19107, USA; 4Sidney Kimmel Cancer Center, Thomas Jefferson University, Philadelphia, PA 19107, USA

**Keywords:** Alzheimer’s disease, Parkinson’s disease, amyotrophic lateral sclerosis, biomarkers

## Abstract

Neurological disorders are the leading cause of cognitive and physical disability worldwide, affecting 15% of the global population. Due to the demographics of aging, the prevalence of neurological disorders, including neurodegenerative diseases, will double over the next two decades. Unfortunately, while available therapies provide symptomatic relief for cognitive and motor impairment, there is an urgent unmet need to develop disease-modifying therapies that slow the rate of pathological progression. In that context, biomarkers could identify at-risk and prodromal patients, monitor disease progression, track responses to therapy, and parse the causality of molecular events to identify novel targets for further clinical investigation. Thus, identifying biomarkers that discriminate between diseases and reflect specific stages of pathology would catalyze the discovery and development of therapeutic targets. This review will describe the prevalence, known mechanisms, ongoing or recently concluded therapeutic clinical trials, and biomarkers of three of the most prevalent neurodegenerative diseases, including Alzheimer’s disease (AD), amyotrophic lateral sclerosis (ALS), and Parkinson’s disease (PD).

## 1. Introduction

Neurodegenerative diseases are debilitating pathologies that erode patients’ cognitive health and physical abilities, leading to a cascading decline in autonomy and quality of life. Neurological disorders, including neurodegenerative diseases, currently affect 15% of the global population, and are expected to double over the next two decades as the worldwide population continues to age [1]. Despite their prevalence, there are few disease-modifying therapies available to treat neurodegenerative diseases, and many obstacles impede their therapeutic development. A major impediment to developing effective disease-modifying therapy is the insidious nature of neurodegenerative diseases, which can progress gradually over the course of years or decades before canonical symptoms of cognitive or physical decline manifest. Unfortunately, by the time most patients are diagnosed with neurodegeneration, many pathological changes have typically occurred, some of which drive and some of which result from the disease. This muddled patchwork without a clear sequence of progression poses a major challenge in discriminating innocent cellular bystanders from toxic drivers of disease, hampering the development of disease-modifying agents. Therefore, to identify paradigm-shifting therapeutic targets, start treatment as early as possible in prodromal and at-risk patients, and confirm the efficacy of therapy in patients, reliable biomarkers reflecting specific underlying processes in distinct stages of disease must be established.

A comprehensive panel of “dry” and “wet” biomarkers comprised of measurements from multiple biological sources will likely prove the most beneficial to patients. Imaging, typically via positron-emission tomography (PET) or single-photon emission computed tomography (SPECT), is a powerful “dry” tool used to identify regional changes in the brain [2,3]. While helpful in differentially diagnosing patients, imaging can be expensive, has limited clinical availability, and exposes patients who require repeated scans to increasing levels of radioactivity. An additional approach is analyzing “wet” biomarkers, which reflect pathological changes in proteins present in biofluid. These markers generally tradeoff between specificity to disease and invasiveness to patients, giving each measurement inherent strengths and weaknesses. Cerebrospinal fluid (CSF), for example, is relatively concentrated with brain-derived markers and has limited contamination by peripheral proteins. Despite these benefits, lumbar puncture is invasive and has been associated with a wide range of deleterious effects that may discourage patients from undergoing routine CSF samplings [4]. Blood draws and the subsequent purification of serum and plasma are far more accessible, affordable, and attractive to patients, but typically have a narrower dynamic range of markers than CSF [5]. Furthermore, subtle changes in blood-based biomarkers may be obscured by peripheral proteins unrelated to neurodegenerative pathology, although research is ongoing to discriminate neural-derived from non-neural circulating proteins [5]. Finally, urine has been widely examined as an accessible biofluid that patients can supply from the comfort of their own homes. Despite the overall lack of specificity of urinary biomarkers, urine may supplement more specific CSF- and blood-based markers to track general ongoing neurodegeneration and inflammation in response to therapy with greater frequency.

The hunt for specific biomarkers of pathology goes hand in glove with the search for disease-modifying therapies. As new medication becomes available to patients suffering from neurodegeneration, specific biomarkers that reflect ongoing pathology would greatly help to monitor individual responses to treatment. Furthermore, discovering biomarkers novel to specific diseases or to organ systems with previously unrecognized roles in pathology could elucidate mechanisms of disease and identify important therapeutic targets. As pathophysiological mechanisms, biomarkers, and therapeutic approaches are tightly interwoven, this review will focus on each of these threads while providing an update on recent translational research in AD, ALS, and PD. Although there are many general markers of neurodegeneration and neuroinflammation, this review will focus on the most specific, reliable, or novel biomarkers within each pathology.

## 2. Alzheimer’s Disease

AD is the most common neurodegenerative disorder in the aging US population, affecting over 2% of adults over 65 years of age [6]. Clinically, early AD is characterized by short-term memory loss, which gradually progresses to a long-term decline in memory, cognitive function, behavior, and social skills [2,6,7,8]. On a molecular level, AD is characterized by an accumulation of amyloid beta (Aβ) plaques, the processing product of amyloid precursor protein (APP) and neurofibrillary tangles (NFTs) comprised of tau protein, in and surrounding neurons throughout multiple brain regions [2,7,8]. While the precise function of APP remains undefined, it may play a role in binding cells and proteins and guiding the migration of neurons during development [9]. During homeostasis, β- and γ-secretases cleave APP into amyloid peptides, primarily Aβ_40_ and Aβ_42_, which are further cleaved by Aβ-degrading enzymes (ADEs) and cleared via microglia and the glymphatic system [2,8]. In AD, however, Aβ peptides accumulate via overproduction or reduced clearance mechanisms, leading to peptide aggregation and the formation of Aβ plaques. These plaques trigger an inflammatory phenotype in microglia, further hindering peptide clearance [2,7,8]. While isolated Aβ plaques are directly toxic to neurons and synapses, they also induce the hyperphosphorylation of tau, a protein that stabilizes neuronal microtubules in homeostasis [2,7,8]. Upon pathologic hyperphosphorylation, tau aggregates accumulate within and are secreted from neurons, building NFTs in neuronal cytoplasm and synapses [2,7,8]. Together, tau tangles and Aβ plaques impede synaptic transmission between neurons and trigger neuroinflammation, ultimately causing neurodegeneration [2,7,8]. The majority of AD is sporadic, but mutations in *APOE*, particularly in the *APOE* ε4 allele, encoding a lipid carrier protein, *APP, PSEN1,* and *PSEN2* (which encode γ-secretase regulators), increase the individual likelihood of developing AD, probably through the impaired clearance of Aβ plaques [2,7,8]. An overview of common molecular mechanisms underlying AD and their resulting established biomarkers is displayed in Figure 1.

### 2.1. Current Prognostic and Diagnostic Indicators for AD

#### 2.1.1. PET Scans

Imaging tools remain the most reliable readouts of AD clinical trials results [2]. Three tracers are FDA-approved for Aβ-plaque imaging, including [18F]-florbetapir, [18F]-flutemetamol, and [18F]-florbetaben [2]. However, as 40% of elderly adults over the age of 90 have Aβ plaques regardless of cognitive impairment, positive Aβ-plaque results should be confirmed with orthogonal approaches [7]. To that end, [18F]-flortaucipir is a tau-specific marker that has been FDA-approved for use in PET imaging. A positive tau PET is more specific than a positive Aβ plaque for diagnosing AD; cognitively typical patients who have positive Aβ plaques and tau PETs are 20x and 40x more likely to develop mild cognitive impairment (MCI) and dementia, respectively. However, a negative tau PET, even in the presence of a positive Aβ-plaque PET, predicts a low likelihood of cognitive impairment [10].

#### 2.1.2. Aβ_42_/Aβ_40_

Research into fluid-based biomarkers in AD has focused primarily on patient CSF and blood. In AD, the APP-cleavage product Aβ_40_ is more readily cleared from the brain parenchyma than is Aβ_42_, the main initiator of Aβ-plaque formation in pathology [11]. Therefore, measuring the ratio of Aβ_42_/Aβ_40_ offers insight into impaired Aβ clearance while accounting for differential APP-processing rates between individuals [11]. A reduced ratio of Aβ_42_/Aβ_40_ is extremely well aligned with Aβ PET positivity, and a decline in CSF levels of Aβ_42_ can predict AD up to 25 years prior to symptom onset [11,12]. Aβ_42_/Aβ_40_ is decreased by 50% in the CSF and 14–20% in the plasma, possibly due to amyloid secreted from non-cerebral tissue such as platelets, binding to blood proteins, and liver metabolism [2,13,14,15]. Despite the comparatively modest decrease in plasma Aβ_42_/Aβ_40_, the relative noninvasiveness of drawing patient blood has encouraged the continued measuring of circulating biomarkers. IP- and LC-MS analyses of plasma Aβ_42_/Aβ_40_ predict Aβ PET positivity with an accuracy of 82–97% [13,16]. Analyses of Aβ_42_/Aβ_40_ plasma levels using single molecule arrays (Simoa), a practical and inexpensive approach for clinics, revealed that the immunoassay detection of plasma Aβ_42_/Aβ_40_ is markedly less precise, but can predict Aβ PET positivity with an accuracy of 62–79% [13]. The Aβ peptide has also been detected in AD patient and AD mouse-model olfactory and buccal epithelium, potentially opening the door to even less invasive sampling methods for early diagnosis [17,18].

#### 2.1.3. Phosphorylated Tau

As hyperphosphorylated tau is secreted from neurons in response to Aβ plaques, total tau (t-tau) and phosphorylated tau (p-tau) levels increase in both the CSF and plasma of AD patients [2,7,8]. T-tau is a less attractive candidate for a fluid-based AD biomarker; despite its half-life of 20 days in CSF, its half-life in blood is only 10 h, limiting its potential to ultimately become an accessible and reliable analyte [19]. Furthermore, there is a high degree of overlapping t-tau levels between AD patients and healthy aging individuals, limiting its specificity and diagnostic capability [2]. P-tau, on the other hand, is being explored for its potential as an AD biomarker. Unlike t-tau, CSF and plasma p-tau differentiate between AD and other similarly presenting clinical diagnoses, such as frontotemporal lobe degeneration (FTLD), with a sensitivity and specificity of 70–85% [20,21]. Tau has over 40 phosphorylation sites, and three forms of p-tau, p-tau181, p-tau217, and p-tau231 are twice as high in the plasma of AD patients compared to healthy individuals. Among these three, p-tau181 is increased 3.5-fold in AD patients and accurately predicts a positive Aβ PET prior to clinical symptoms [20]. As in the plasma, CSF p-tau181 is predictive of AD pathology and can identify patients up to 10 years prior to symptom onset [12]. P-tau181 strongly correlates with p-tau PET [22], and can reliably distinguish between AD and other causes of dementia, including other tauopathies [20,23,24]. P-tau217 may be even more accurate than p-tau181 at predicting Aβ-PET positivity, correlating with tau-PET [25], and correctly differentiating AD from other forms of dementia in CSF and plasma in up to 96% of cases [26,27,28]. Additionally, plasma p-tau217 becomes abnormal prior to tau PET in autosomal-dominant AD [29], underscoring its potential to identify at-risk patients and help guide preventative therapy. Plasma p-tau231 is another promising fluid biomarker; among a panel of biomarkers including p-tau181, p-tau217, p-tau231, glial fibrillary acid protein (GFAP), neurofilament light (NfL) and Aβ_42/40_, p-tau217 and p-tau231 most closely correlated with reduced CSF levels of Aβ_42/40_. However, p-tau217 was the most sensitive to small increases in Aβ PET positivity, even prior to overt plaque formation and clinical symptom onset [30]. Importantly, despite its close association with Aβ plaques, p-tau231 does not correlate with tau tangle load, suggesting a novel mechanism for its rising plasma levels in AD [31]. While altered levels in blood compartments have been more thoroughly studied, t-tau and p-tau are also upregulated in AD lymphocytes, and p-tau overexpression has been characterized in AD platelets and olfactory and buccal epithelia, facilitating their potential as accessible biomarkers [17,32].

#### 2.1.4. Neurofilament Light

Neurofilaments are scaffolding proteins that are found throughout neurons, but are most highly concentrated in axons where they promote axonal growth and maintain structural stability [33]. Though not specific to AD, CSF and plasma NfL levels significantly increase in later stages of AD progression as neurons degenerate and axonal proteins are shed [13]. NfL increases more in other differential diagnoses, including progressive supranuclear palsy (PSP), corticobasal syndrome (CBS), and FTLD [20], and may hold more promise as a biomarker for other neurodegenerative diseases, as described below. While NfL has limited potential as a diagnostic indicator for AD, it may help to track responses to disease-modifying therapy.

### 2.2. Biomarker Use and Misuse in AD Therapies and Clinical Trials

Until relatively recently, approved AD therapies aimed to ameliorate the cognitive impairment caused by the disease but did not address the underlying drivers of the pathology. However, in 2021 and 2023, the FDA approved aducanumab and lecanemab, two monoclonal antibodies that target Aβ plaques, for use in patients with MCI or AD [34,35]. These drugs have a modest positive impact on AD biomarkers, as determined by PET imaging of Aβ plaque and tau tangle load, CSF sampling of Aβ_42_ and p-tau181, and plasma levels of p-tau181 [34,35]. However, these promising results alone do not guarantee improvements in patient outcomes, and a recent controversy in AD drug development underscores the potential misalignment between changes in key pathological biomarkers and clinical results. Despite widespread concern among researchers that aducanumab treatment did not cause clinical improvements, the FDA greenlit the drug for AD patient use via accelerated approval based on encouraging biomarker data [36]. Accelerated approval is reserved for drugs aimed to treat diseases with limited therapeutic options and allows grantees to provide post-approval clinical outcome data. Unfortunately, after aducanumab became available to AD patients, the manufacturer failed to find a significant deceleration in cognitive decline and subsequently withdrew the drug from the market in early February 2024 [36,37,38]. Lecanemab, manufactured by the same company, was also granted accelerated FDA approval, raising concerns about its efficacy in the shadow of aducanumab’s clinical limitations. Fortunately, recent clinical trial results have revealed that lecanemab slows cognitive decline by 27% over 18 months of treatment in AD patients, possibly due to its greater efficacy in clearing Aβ plaques [36]. While further studies are required to investigate the impact of the drug over a longer period of time, this positive finding has helped to restore a degree of confidence in the still-controversial accelerated-approval process. However, the precarious journey of aducanumab may serve as a cautionary tale that emphasizes the importance of considering encouraging biomarker data in the context of patient outcomes.

The clinical success of lecanemab supports the pharmaceutical approach of clearing Aβ plaques to treat AD. Currently, there are 73 ongoing clinical trials for AD patients in the U.S., and most focus on targeting Aβ plaques with monoclonal antibodies or binding to Aβ peptides to prevent plaque formation (publicly available to view on the National Institute on Aging website). Most of these trials rely on cognitive testing and Aβ PET imaging as their endpoints to determine efficacy and would likely benefit from the additional monitoring of fluid biomarkers as described above.

## 3. Amyotrophic Lateral Sclerosis

ALS is a devastating disease that causes the degeneration of upper (bulbar) and lower (spinal) motor neurons, leading to rigid and flaccid paralysis, respectively [39,40]. While relatively rare (affecting 5.5 per 100,000 persons), ALS is the leading cause of motor neuron degenerative pathology in adults, and currently there are no available therapies to slow disease progression [39,40]. From diagnosis, ALS patients have an average survival of 3–5 years, and death most frequently reflects respiratory failure as motor neurons innervating the diaphragm fail [39,40]. Beyond neuromuscular symptoms, 50% of ALS patients experience cognitive or behavioral changes, and frontotemporal dementia (FTD) is diagnosed in 5–25% of ALS cases [39,40]. The etiology of ALS is complex and multifaceted, posing a challenge to determining the causality of cellular events. Ten percent of ALS is familial, and mutations in *SOD1* (encoding an antioxidant), *FUS* (encoding a regulator of translation), *C9ORF72* (encoding a regulator of autophagy), and *TARDBP* (encoding a regulator of translation) represent high risks for developing the disease [39,40]. The other 90% of ALS is sporadic, caused by an intricate interplay between undefined genetic susceptibility and environmental events. Regardless of origin, familial and sporadic ALS disease development converge on an early neurodegenerative pathway and are clinically indistinguishable at the time of diagnosis [39,40]. On a cellular level, 97% of ALS patients have abnormal ubiquitinated protein aggregates of TAR DNA-binding protein 43 (TDP43) in neuronal cytoplasm [41,42,43]. This mislocalization of TDP43 from the nucleus to the cytoplasm has been associated with impaired ribosomal function, altered RNA translation, aberrant splicing, and cryptic peptide formation [42,43,44]. These cryptic peptides can insert themselves into the sequence of what would otherwise be properly translated proteins, impeding normal function [42,43,44]. Beyond impaired TDP43 localization, motor neurons in ALS patients have disrupted mitochondrial respiration, elevated levels of oxidative stress, and increased glutamatergic neurotoxicity [44,45]. A combination of established and novel changes occurring during ALS with the resulting biomarkers across multiple sample types are shown in Figure 2.

### 3.1. Current Prognostic and Diagnostic Indicators for ALS

#### 3.1.1. Neurofilaments

Due to the rapid progression of disease from symptom onset to patient death, there is an urgent need to develop early detectors of ALS and widen the therapeutic window for future disease-modifying treatments to intervene in pathological progression. Unfortunately, the multifactorial nature of ALS pathology has hampered the search for a disease-specific biomarker that encompasses a high enough proportion of ALS patients to be practically useful [46]. Rather, a panel of biomarkers, including general readouts of neurodegeneration, is more likely to hold the most promise for presymptomatic or early disease detection. Neurofilaments are the leading candidates in the search for ALS biomarkers and are currently used in clinical trials as endpoint readouts of therapeutic efficacy with mixed results. Although neurofilaments are elevated across neurodegenerative disease, the CSF and plasma levels of NfL and the phosphorylated neurofilament heavy chain (p-NfH) are highest in ALS compared to healthy controls, neurodegenerative disorders such as AD, FTD, and corticobasal syndrome (CBS), and other motor-nerve disorders [47,48]. Importantly, CSF and plasma levels of NfL and p-NfH correlate with ALS severity and progression [48,49,50], and plasma NfL increases up to one year prior to symptomatic onset [51]. In parallel to this observation, CSF and plasma NfL and p-NfH levels are highest in patients with a bulbar, rather than spinal, onset of disease, associated with a more aggressive progression of ALS pathology [48].

#### 3.1.2. TDP43

Phosphorylated, ubiquitinated cytoplasmic inclusions of DNA-binding protein TDP43 within cortical motor neurons are a histological hallmark of ALS, and plasma levels of TDP43 are inversely correlated with disease progression, suggesting a deficiency in clearing pathological aggregates from the brain [52]. Despite the prevalence of aggregated TDP43 in pathological motor neurons, the impact of mislocalization from the nucleus to the cytoplasm on disease progression remains an emerging area of ALS research. The loss of TDP43 from the nucleus causes DNA damage, errors in splicing, somatic mutations, and fusion proteins, leading to a wide range of downstream effects [53]. Aberrant TDP43-induced DNA damage is likely responsible for the observed increased activity of p53 within ALS motor neurons, providing a direct link between TDP43 mislocalization and apoptosis in pathology [53]. Beyond triggering DNA damage-response pathways, TDP43 dysregulation also has been associated with deleterious effects driven by mis-spliced proteins and the loss of their homeostatic functions. Stathmin-2 (STMN2), for example, is a protein partially regulated by TDP43 that repairs damaged axons and guides neural regrowth following damage. This protein is frequently mis-spliced in ALS patients, which coincides with reduced levels of canonical STMN2 and increased vulnerability to neurodegeneration [44,54,55].

While the pathological significance of dysregulated TDP43 and its malignant mis-spliced products in ALS remain under investigation, the potentially benign by-products of TDP43 mislocalization and aggregation are currently being studied as possible biomarkers for diagnosing preclinical disease. In the past year, two research groups have identified specific cryptic exons, products of splicing errors, as signatures of TDP43 pathology prevalent in ALS patients [56,57]. Critically, rising levels of cryptic exon HDGFL2 in the CSF of high-risk, yet asymptomatic, ALS patients precede profound axonal degeneration, as determined by sharply increased CSF levels of neurofilament [56]. This finding may be a key foothold in determining the sequence of molecular events that precede neuromuscular atrophy in ALS. As proposed ALS biomarkers have thus far been limited to those reflecting ongoing neurodegeneration and neuroinflammation, TDP43-specific cryptic exons are among the most exciting and hopeful findings in the field for identifying preclinical pathology and slowing disease progression. Furthermore, their discovery suggests the translational utility of gene therapy to deliver functional TDP43 to ALS patients prior to, and early in, symptomatic onset, although further research is necessary to determine the feasibility of this approach.

#### 3.1.3. Chitinases

Chitinases are glycosyl hydrolase enzymes without a defined function in mammals. Despite limited knowledge about the significance of their biological activity, chitinases have increasingly gained recognition as markers of neuroinflammation and degeneration in neurological diseases [58,59]. CSF levels of chitinase-1 (CHIT1), chitinase-3-like-protein-1 (CHIT3L1), and chitinase-3-like-protein-2 (CHIT3L2) are upregulated in ALS CSF as compared to healthy and disease controls and correlate with disease aggressiveness [49,60,61]. Importantly, these markers rise prior to symptom onset in at-risk individuals, emphasizing the potential for CSF levels of chitinases to predict phenoconversion (the tipping point between prodromal disease and symptom onset) [62]. Despite the consistent elevation of CHIT1, CHIT3L1, and CHIT3L2 in CSF, ALS plasma levels of chitinases are far more variable with little to no correlation to pathology [58,61], limiting the feasibility of the long-term tracking of disease progression via noninvasive methods.

#### 3.1.4. Urinary Markers: p75^ECD^ and Neopterin

The extracellular domain of p75 (p75^ECD^) is upregulated on the surface of Schwann cells and apoptotic motor neurons. Upon surface expression, p75^ECD^ is cleaved and secreted into the bloodstream and is ultimately detectable in urine, where its presence reflects motor neuron injury and its levels correlate with disease progression [63,64]. Another urinary marker, neopterin, is released from monocytic cells, including macrophages and microglia, upon exposure to interferon-gamma (IFNγ). Although widely considered to be a general marker of neuroinflammation, urinary levels of neopterin are significantly higher in ALS patients compared to healthy controls, MS patients, or patients with other neurodegenerative diseases [65,66]. To date, attempts to correlate urinary levels of neopterin with disease progression have yielded mixed results, and further investigation is required to demonstrate the utility of neopterin in monitoring pathology and responses to therapy [65,66]. However, the significance of developing a panel of biomarkers for monitoring disease progression and responses to therapy that include urinary proteins such as p75^ECD^ and neopterin should not be minimized. Lumbar punctures for obtaining CSF samples commonly cause headaches and back pain for patients, and rarely can cause more serious adverse events [4]. While blood samples are far less invasive to procure, urine samples can be obtained from the comfort of patients’ homes and may therefore offer practical and continuous monitoring of disease stages to complement less frequent measurements taken from more informative biofluids.

### 3.2. Biomarker Use in ALS Therapies and Clinical Trials

Alongside functional patient outcomes, NfL has been widely used as an endpoint readout in a wide range of ALS clinical trials to determine responses to therapy, albeit with mixed results. Four therapies are currently FDA-approved for mitigating the symptoms of ALS, with modest improvements in patient survival. Riluzole, an inhibitor of glutamatergic release and transmission, reduces neuronal excitotoxicity, improves patient symptoms measured by limb and respiratory function, and extends the survival of ALS patients by 3–19 months [39]. However, a recent study reveals that riluzole does not reduce serum NfL levels in ALS patients [67], calling into question either the ability of riluzole to slow neurodegeneration or the utility of NfL as a reliable biomarker to track responses to therapy. Edaravone, a reactive-oxygen-species (ROS) scavenger, improves patients’ functional scores, but its effects on survival are not significant [68,69]. NfL has not been measured in patients having received edaravone, so the impact of this therapy on quantifiable neurodegeneration remains undefined. AMX0035 is a fixed-dose combination of taurursodiol and sodium phenylbutyrate that limits mitochondrial dysfunction and endoplasmic reticulum (ER) stress and is currently approved for ALS patients. In a clinical trial, AMX0035 extended survival by 6.5 months on average and improved patients’ functional outcomes [70,71]. Despite these benefits, AMX0035 failed to lower measured levels of NfL, adding to the uncertainty of relying on NfL as a reflection of therapy response [72,73]. Toferson, however, is an SOD1 antisense oligonucleotide that is under consideration for treating ALS patients with SOD1 mutations. In a 28-week clinical trial, toferson failed to improve functional patient outcomes, but significantly reduced CSF and plasma levels of NfL, suggesting a long-term potential to slow neurodegeneration [74,75]. Tofersen therefore is being evaluated in presymptomatic ALS patients with SOD1 mutations, with the expectation that a wider therapeutic window will give more of an opportunity for reductions in NfL, presumably reflecting slower rates of neurodegeneration that translate to functional patient outcomes. While the degree of disease modification of current treatments and the utility of NfL in clinical trials continue to be investigated, the field of ALS-therapy development would greatly benefit from the establishment of additional novel biomarkers of early or preclinical pathology.

## 4. Parkinson’s Disease

PD is characterized by a profound loss of dopaminergic (DA) neurons within the substantia nigra pars compacta (SNpc), the brain structure responsible for the release of DA into the striatum to ultimately control voluntary movement [76,77]. PD patients suffer from motor deficits such as cogwheel rigidity, resting tremor, shuffling gait, and small, cramped handwriting [76,77]. PD occurs in about 1% of the US population over 60 years of age and is more prevalent among men [76,77]. Multiple genetic mutations and environmental causes have been linked to PD development, including mutations in *GBA* (encoding glucocerebrosidase, a mediator of autophagy), *SNCA* (encoding α-synuclein), *LRRK2* (encoding leucine-rich repeat kinase 1, a mediator of autophagy), and *PINK1* (a mediator of mitophagy), among others [77]. These observations have helped to identify the role of impaired autophagy, dysfunctional mitochondrial turnover, increased oxidative stress, and dysregulated autoimmunity in the pathogenesis of PD [77]. Viral infections such as influenza and COVID19, exposure to industrial pesticides, and increased contact with heavy metals have also been tied to the increased vulnerability of DA neurons to degeneration later in life, further supporting the impacts of the immune system, oxidative stress, and mitochondrial dysfunction on PD pathogenesis [78,79,80]. Beyond DA neurodegeneration, most PD patients suffer from GI-related symptoms, including dysphagia, delayed colonic transit time, and constipation, and these symptoms can develop decades prior to motor deficits [81,82].

Lewy bodies, comprised of aggregated pathological α-synuclein, are the histological hallmark of PD pathology, and have been demonstrated to trigger a chain of misfolding in healthy proteins, migrate from cell to cell in a prion-like fashion, and promote neurodegeneration and neuroinflammation [83,84]. There are multiple potential sites of initial pathological protein aggregation; the landmark Braak hypothesis posits that α-synuclein misfolding begins in the gut, migrates up the vagus nerve to the olfactory bulb, and spreads throughout DA neural circuitry to reach DA neurons in the SNpc [83,85]. In support of this theory, Lewy bodies have been identified in presymptomatic PD-patient intestine, and PD patients frequently report GI symptoms and anosmia decades prior to developing motor deficits [81,82,86]. Ultimately, a cellular overload of α-synuclein may be one among many causes of oxidative stress and impaired mitochondrial function, which prevents DA neurons from producing sufficient ATP and triggers ROS-dependent apoptotic cascades [87]. Multiple intrinsic qualities render SNpc DA neurons uniquely vulnerable to elevated levels of cellular stress and apoptosis. DA production and metabolism create products that autoxidize, raising ROS levels in homeostasis [88]. Furthermore, SNpc DA neurons are particularly large and densely arborized, leading to a high energy demand [89]. Furthermore, unlike DA neurons in the neighboring ventral tegmental area (VTA), SNpc DA neurons lack high levels of antioxidants and are more vulnerable to cell death [90,91,92]. Therefore, even relatively small reductions in mitochondrial energy production or increases in oxidative stress levels can tip SNpc DA neurons toward apoptosis and neurodegeneration. Established molecular changes and resulting biomarkers measured across different samples in PD patients are displayed in Figure 3.

### 4.1. Current Prognostic and Diagnostic Indicators for PD

#### 4.1.1. Imaging

Due to the high degree of overlap in DA circuitry between the SNpc and the striatum, PD can progress insidiously for years without manifesting clinical symptoms. However, once 50–90% of DA neurons have been lost from the SNpc, patients experience initial motor deficits that are typically confirmed as PD-induced by imaging reductions in DA neurons in the SNpc via DAT-SPECT [3]. Unfortunately, for DAT-SPECT to work effectively, patients must lose a profound number of DA neurons prior to receiving a confirmatory diagnosis, narrowing the window of future disease-modifying therapies to slow or prevent neurodegeneration. Alternative methods of imaging pathological changes in vulnerable populations may focus on the altered expression of surface markers that reflect DA neuron stress prior to cell death. Guanylyl cyclase C (GUCY2C), a surface receptor expressed in the intestine and specific neural pathways, is upregulated on the surface of SNpc DA neurons in mice having received MPTP, a mitochondrial toxin that induces neurodegeneration [93]. Importantly, GUCY2C RNA levels also are upregulated in the DA neurons of PD patients, although GUCY2C protein expression and surface localization in PD need to be confirmed before assessing its clinical utility [93]. Ultimately, radiotracer detection of surface proteins that correlate with pathological changes may allow PET imaging to identify high-risk individuals on the precipice of DA neurodegeneration, providing an opportunity to intervene in disease progression prior to developing irreversible motor deficits.

#### 4.1.2. Alpha-Synuclein

Monomeric α-synuclein is located within neuronal nuclei, mitochondria, cytoplasm, and synaptic terminals, as well as in other non-neuronal cells [94]. Although the functional role that monomeric α-synuclein plays in homeostasis is not fully understood, oligomeric and aggregated α-synuclein is widely recognized as a prominent histological hallmark and potential driver of PD pathology [94]. Alpha-synuclein is the leading potential new biomarker for PD, and there have been many recent technological developments to improve the accuracy, accessibility, and speed of quantifying levels of α-synuclein from patient samples to diagnose disease. Most recent studies have used an array of assays to determine the ability of sampled α-synuclein (the “seed”) to trigger a chain of misfolding in healthy proteins and can be grouped together under the umbrella of seed-amplification assays (SAAs) [95]. A thorough analysis of 1100 participants in the Parkinson’s Progression Markers Initiative (PPMI) recently revealed the high accuracy of CSF-derived α-synuclein SAA in diagnosing PD and identifying prodromal patients with a sensitivity of 93% and 86%, respectively [96]. Despite the high degree of accuracy that SAAs offer, these results are widely considered to be binary, and are thus better equipped to diagnose PD patients than to monitor responses to therapy. However, the immunoassay detection of pathologic α-synuclein produces continuous data, which may provide complementary information to accurately track responses to treatment. ELISA analysis of CSF-derived α-synuclein oligomers and early and highly pathologic forms of α-synuclein aggregates accurately discriminates PD patients from healthy controls and, when combined with SAA, may offer a higher degree of accuracy in diagnosis than SAA results alone [97]. Furthermore, immunoassay rapidly detects α-synuclein oligomer load, and the results correlate with motor impairment [97].

Despite the accuracy of CSF-based α-synuclein analysis, the invasiveness of CSF sampling may discourage patients with mild to moderate prodromal symptoms from undergoing lumbar puncture for testing [4]. Thus, the field would greatly benefit from a blood-based biomarker. Early attempts to quantify circulating α-synuclein in the blood of PD patients yielded mixed results and were likely confounded by contaminating levels of α-synuclein released from peripheral sources [5]. However, the discovery of neuron-derived α-synuclein within circulating extracellular vesicles (EVs) has opened the door to accurately quantifying pathological α-synuclein using minimally invasive sampling methods. Elevated α-synuclein within neuron-derived L1 cell adhesion molecule-positive extracellular vesicles (LEV) distinguishes prodromal PD patients from healthy controls with 85–91% sensitivity, discriminates between PD and tauopathies with 98% sensitivity, and accurately identifies early-PD patients with minimal motor impairment [98,99,100]. Like CSF-derived α-synuclein, EV-α-synuclein isolated from PD-patient blood yields a positive SAA result [101]. Furthermore, LEV-α-synuclein levels tightly correlate with the degree of motor impairment, emphasizing the potential to use these measurements as a readout of responses to future disease-modifying therapies [98]. Within individual PD patients, longitudinal increases over time, rather than absolute levels, of LEV-α-synuclein are more tightly associated with the progression of motor impairment, underscoring the advantage of a blood-based biomarker to track disease stage [100].

The frontier of minimally invasive approaches to use α-synuclein as a biomarker for PD is continually advancing. Oligomerized α-synuclein is elevated in EVs isolated from PD patient saliva, as compared to controls, and can identify PD patients with 92% sensitivity [102]. Furthermore, skin scrapings from postmortem PD patients yield positive α-synuclein SAA with 99% sensitivity [103]. Alpha-synuclein SAA performed on combined skin scraping and nasal brushing accurately identifies 95% of PD patients, although PD patients will frequently have negative nasal-brushing results and sometimes have negative skin-scraping results [104]. This variation in pathological α-synuclein load may preclude the noninvasive sampling of skin and the nasal epithelium from being reliable diagnostic indicators. However, the varied distribution of this widely recognized biomarker from specific tissues may help to illuminate multiple peripheral pathological pathways that converge on DA neurodegeneration. Interestingly, α-synuclein is also elevated in the stool of prodromal, but not symptomatic, PD patients [105], which may suggest a role for pathological α-synuclein in the onset of gastrointestinal symptoms prior to motor deficits [106].

#### 4.1.3. MIRO1

Mitochondrial Rho GTPase (MIRO1) is a mitochondrial surface protein that recruits microtubules to assist in mitochondrial motility [107]. Upon damage, mitochondrial membranes are depolarized and MIRO1 is removed from the outer mitochondrial membrane through several mediators including PINK1, PARKIN, and LRRK2 [107], likely arresting mitochondria in preparation for mitophagy. However, in PD, MIRO1 degradation and subsequent mitochondrial turnover is commonly impaired. In 94% of a broad, heterogeneous PD-patient cohort comprised of individuals with identified genetic mutations or sporadic pathology, MIRO1 abnormally persisted on the surface of depolarized mitochondria in skin-derived fibroblasts [108]. MIRO1 degradation also is impaired in fibroblasts isolated from at-risk individuals harboring gene mutations in *GBA1* and *LRRK2* [109]. This landmark discovery from noninvasive skin samples raises the possibility of using MIRO1 persistence as a biomarker in early PD pathogenesis, although more research is required to investigate MIRO1 resistance to degradation in prodromal PD patients. Furthermore, as mitophagy is a commonly dysregulated process in PD patients and may underlie DA neuron loss, these findings introduce MIRO1 as a novel potential therapeutic target worthy of additional investigation.

While identifying a biomarker for dysfunctional mitophagy remains an emerging frontier for PD, this potential development also holds far-reaching implications for the neurodegenerative field more broadly. Mitochondrial dysfunction and resulting oxidative stress is nearly ubiquitous across neurodegenerative disease, and impaired mitochondrial trafficking and altered mitochondrial dynamics have been implicated in AD, ALS, and other neurodegenerative conditions [110]. In AD, postmortem brain samples have significantly lower levels of mitochondrial proteins related to oxidative phosphorylation and reduced translocase of outer membrane (TOM) proteins TOM20 and TOM70 [111,112]. The deleterious relationship between impaired mitochondria and AD pathology may be bidirectional; the infiltration of Aβ into mitochondria impairs mitochondrial energy metabolism and increases oxidative stress, a driver of synaptic dysfunction and neurodegeneration [112,113]. AD patients also have compromised mitochondria in peripheral tissue, suggesting an underlying metabolic dysfunction that may contribute to early plaque formation [114]. In ALS, postmortem and biopsy samples reveal impaired mitochondrial structure, number, localization, and metabolism [115]. Furthermore, motor neurons developed from both sporadic and familial ALS iPSCs have reduced ATP, depolarized mitochondria, impaired oxidative phosphorylation, and elevated oxidative stress [116], and the therapeutic potential of restoring healthy levels of mitochondrial metabolic products is currently under investigation for treating ALS [117]. As a common thread among a wide range of neurodegenerative conditions, determining key markers of mitochondrial dysfunction may prove to be an invaluable screening tool in the early identification of neurodegenerative diseases.

#### 4.1.4. Glucocerebrosidase Activity

*GBA* is frequently mutated in familial PD, accounting for 5–15% of PD cases in the United States [118]. *GBA* encodes the lysosomal hydrolase enzyme glucocerebrosidase (GC) and plays an important role in mediating autophagy and reducing ER stress [118]. Hundreds of mutations in *GBA* have been clinically documented, with both gain and loss of functions observed in aberrant translated proteins [118]. Mutant GC has been implicated in elevating oxidative stress, impairing mitochondrial functions, and promoting α-synuclein aggregation, leading to multiple pathologies, including PD [118]. Compared to healthy individuals and idiopathic-PD patients, *GBA*-associated PD (*GBA*-PD) patients have reduced CSF GC activity, correlating with elevated levels of GC substrate glucosylceramide and reduced levels of its downstream metabolite sphingomyelin in CSF [119,120]. Importantly, elevated CSF ratios of glucosylceramide to sphingomyelin correlate with accelerated cognitive decline [119]. Glucosylceramide also is elevated in *GBA*-PD plasma, which parallels reduced GC activity in *GBA*-PD whole blood [121,122].

#### 4.1.5. Microbiome

Dysregulated gut microbiota have been well documented in PD patients and are associated with SNpc neuroinflammation and neurodegeneration in mouse models of DA neuron loss [123,124]. PD patients have reduced diversity in the microbiome, and multiple bacterial species, including *Geotrichum candidum*, *Linum usitatissimum*, and *Penicillium roqueforti* are differentially expressed between PD patients and healthy individuals [124]. While there is not a standardized microbiome signature that identifies early PD, recent technological developments have advanced the possibility of using the dysregulated gut as a biomarker for PD. Shotgun metagenomic profiling of the microbiome has found that prodromal and recently diagnosed PD patients have reduced intestinal levels of strict anaerobes and determined that profiling stool samples can identify patients with a recent diagnosis of PD with 76% accuracy [125]. Further refinement is necessary to accurately diagnose prodromal PD patients using a microbiome panel. However, since about 80% of PD patients have GI symptoms up to decades prior to motor dysfunction, developing a diagnostic tool that can identify PD pathology through stool sampling would greatly advance the therapeutic window for future disease-modifying treatments.

### 4.2. Biomarker Use in PD Therapies and Clinical Trials

FDA-approved therapies for PD patients address the motor symptoms of the disease and primarily focus on boosting DA production, limiting DA catabolism, or mimicking the effects of DA to stimulate motor neurons [126,127,128]. While suppressing PD symptoms and improving patients’ quality of life is critical, there is still an urgent unmet need to develop disease-modifying therapies to slow the rate of DA neurodegeneration. Furthermore, as a loss of DA neurons directly causes motor impairment, PD biomarkers are necessary to distinguish between interventional therapies that improve motor function by slowing the rate of neurodegeneration from those that provide relief without preventing pathological progression.

#### 4.2.1. Anti-α-Synuclein

Two anti-α-synuclein monoclonal antibodies, cimpanemab and prasinezumab, have recently been tested in PD patients. The antibodies preferentially targeted aggregated polymeric α-synuclein, although prasinezumab had a slight affinity for the monomeric form to limit the pool of available healthy filaments that could be corrupted to become pathological. Unfortunately, both of these antibodies failed to reduce α-synuclein levels in the CSF or plasma and did not slow the progression of motor impairment or the loss of imaged DAT compared to controls [129,130]. Buntanetap, which inhibits the translation of neurotoxic proteins such as α-synuclein, may hold more promise, and has recently been demonstrated to improve motor scores and lower levels of α-synuclein in the CSF in a small cohort of early-PD patients (clinical-trial identifier NCT04524351) [131]. However, DAT-SPECT and longitudinal monitoring are necessary to determine whether there is long-term potential for buntanetap to slow the rate of neurodegeneration.

#### 4.2.2. GBA-Related Targets

As mutations in *GBA* underlie a substantial portion of PD pathology, therapies focused on counteracting the deleterious effects of aberrant GC have been tested in *GBA*-PD patients. Venglustat (GZ667161) inhibits glucosylceramide synthase and was administered to *GBA*-PD patients to reduce the burden of glucosylceramides in the SNpc (clinical trial ID NCT02906020). Unfortunately, despite the reduction in CSF and plasma levels of glucosylceramides, venglustat was found to both worsen motor deficits and have a negligible, and possible worsening, effect on cognitive decline. Although clinical trials determining the therapeutic potential of venglustat in treating other GBA-related disorders such as Gaucher’s disease have continued, the trial studying its utility in *GBA*-PD patients was recently halted.

#### 4.2.3. Microbiome Restoration

Due to the documented role that dysregulated gut microbiota play in promoting neuroinflammation and accelerating DA neurodegeneration, rebalancing aberrant patient microbiomes is an ongoing field of research in PD-therapy development [121,123]. Interestingly, fecal microbiota transplants in patients with mild to moderate PD has improved GI motility and motor scores, although whether these benefits are long-lasting remains to be determined [121,132]. Expanded longitudinal studies with DAT-SPECT imaging in early- and preclinical-PD patients would greatly help to elucidate whether fecal microbiota transplants may slow the progression of DA neurodegeneration.

## 5. Conclusions

As disease-modifying therapies for currently incurable, debilitating neurodegenerative diseases continue to advance, so too must the search for biomarkers specific to unique pathologies. These markers will help to identify at-risk patients to widen and advance treatment windows, differentially diagnose diseases to ensure that patients receive the most effective therapies, and monitor responses to medication. The importance of developing these biomarkers cannot be understated. Beyond the direct clinical value they offer, biomarkers reflect specific stages of disease progression, thus helping to chronicle the sequence of pathological events, uncover novel therapeutic targets, and discriminate correlation from causation. Despite their critical role in monitoring responses to therapy, changes in biomarkers must be considered in the context of patient outcomes to ensure alignment between molecular and clinical improvements. Ultimately, demystifying pathological origins, achieving therapeutic breakthroughs, and identifying biomarkers specific to diseases and stages represent a three-pronged approach that must be combined and harnessed to combat the mounting problem of neurodegeneration among aging demographics.

## 6. Hopeful Directions on the Horizon

Reduced CSF and plasma levels of Aβ_42_/Aβ_40_ and p-tau are strongly correlated with different stages of AD development. They are effective in identifying prodromal patients, offering therapeutic targets, and monitoring responses to disease-modifying therapy, and will likely pave the way in validating future therapies;Although NfL has not been definitively linked to current disease-modifying therapies for treating ALS, this lack of association must be considered in the context of the narrow treatment window that patients typically have from diagnosis until death. More research is necessary to determine whether currently available ALS treatments slow the rate of neurodegeneration, and whether this correlates with NfL levels over time;New biomarkers that reflect TDP43 pathology offer fresh hope in identifying early disease and using gene therapy to restore healthy splicing regulation. Research into the potential of TDP43 gene therapy should investigate the ability to delay or prevent neurodegeneration as determined through motor scores, cryptic peptide levels, and NfL levels;Alpha-synuclein is a well-established biomarker for predicting, diagnosing, and tracking the progression of PD severity. However, the lack of disease-modifying therapies available to PD patients precludes the ability to use biomarkers to monitor responses to treatment. Further research is necessary to characterize early changes in mitophagy and the microbiome in prodromal patients to help identify potential therapeutic targets and establish additional readouts of pathology and responses to treatment.

## Figures and Tables

**Figure 1 biomolecules-14-00398-f001:**
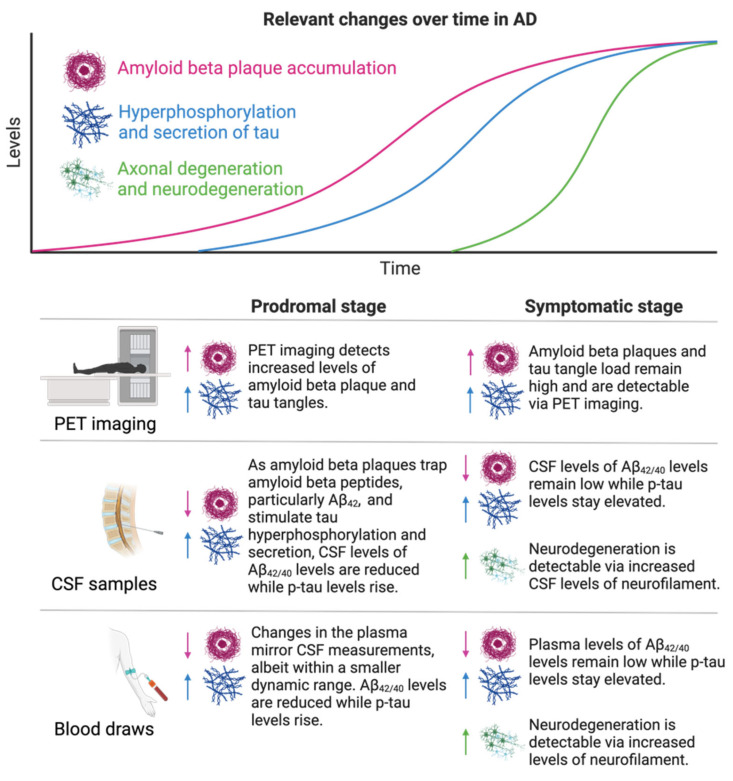
**Relevant changes over time in AD.** AD is characterized by increasing levels of Aβ plaque, followed by hyperphosphorylation and secretion of tau and subsequent neurodegeneration. These changes can be monitored using multiple readouts to help diagnose patients and distinguish between the prodromal (shown in left column) and symptomatic (shown in right column) stages of disease. PET imaging, CSF sampling, and blood draws are most commonly used in ongoing research to asses pathological changes in AD patients over the course of disease progression. Within each specified sampling compartment, arrows pointing up represent elevations in biomarkers that occur during pathology, while arrows pointing down represent declines in these biomarkers.

**Figure 2 biomolecules-14-00398-f002:**
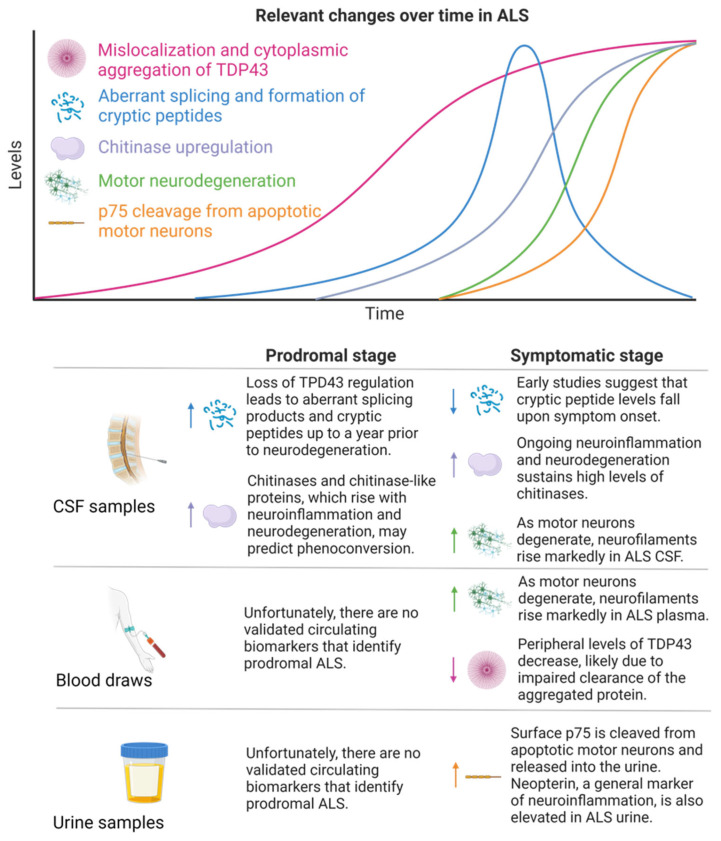
**Relevant changes over time in ALS.** In ALS, abberant localization and aggregation of TDP43 in motor neurons indirectly induces the formation of cryptic peptides. This, along with chitinase upregulation, precedes motor neurodegeneration. As motor neurons degenerate, p75 is cleaved from the apoptotic surfaces and is ultimately detectable in urine. Changes in biomarkers found in ALS patient CSF, blood, and urine are ongoing areas of investigation to identify prodromal (shown in left column) disease and accurately diagnose pathology in the symptomatic (shown in right column) stage. Within each specified sampling compartment, arrows pointing up represent elevations in biomarkers that occur during pathology, while arrows pointing down represent declines in these biomarkers.

**Figure 3 biomolecules-14-00398-f003:**
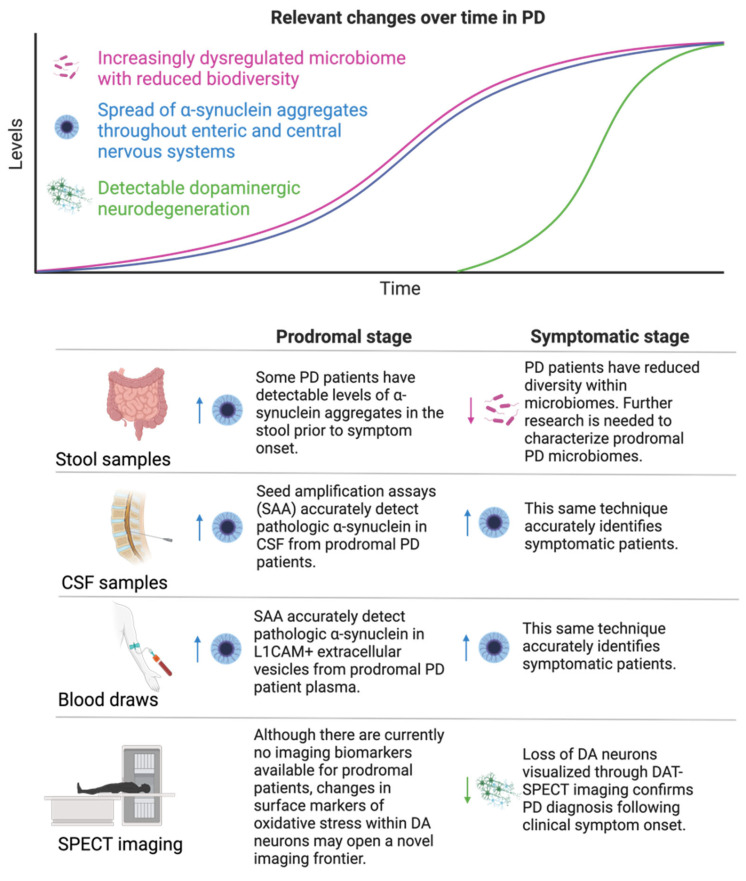
**Relevant changes over time in PD.** PD is characterized by a profound loss of DA neurons within the SNpc, which is diagnosed using confirmatory SPECT imaging. Although there are currently no approved biomarkers to identify PD patients in earlier stages of disease, declines in PD patient microbiome diversity and elevations in pathological forms of α-synuclein prior to symptom onset have been well documented. Sampling PD patient stool, CSF, and blood holds promise for identifying prodromal (shown in left column) patients and monitoring disease progression during the symptomatic (shown in right column) stage of pathology. Within each specified sampling compartment, arrows pointing up represent elevations in biomarkers that occur during pathology, while arrows pointing down represent declines in these biomarkers.

## Data Availability

No new data were created or analyzed in this study. Data sharing is not applicable to this article.

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
