# Peer review of "Biomarkers for Managing Neurodegenerative Diseases"

_biomolecules, 2024, doi:10.3390/biom14040398_

Round 1
Reviewer 1 Report
Comments and Suggestions for Authors
The paper as a whole represents a fairly complete overview of the current state of the study of biomarkers in neurodegenerative diseases and is of interest to a wide range of physicians and biologists.
However, there are some omissions in the article that I would like to draw the attention of the authors and editor.
For an exhaustive completeness of the review and the formation of modern ideas in the reader about this important socially significant pathology, I recommend supplementing the article with the following information:
1) insert a small separate section devoted to the inclusion of neurodegenerative pathology in the group of mitochondrial diseases and highlight the role of mitochondrial proteins (especially TOM20 and TOM70) in the signaling mechanisms of mitochondrial dysfunction involved in the process of neurodegeneration (the works of the group of Prof. J. Hernandez-Yago from Valencia);
2) to section 2.1.3. provide information that blood lymphocytes and buccal epithelium, in which the expression of phosphorylated tau protein and other molecules associated with Alzheimer's disease has been verified, can be considered as a convenient object for life-time diagnosis and monitoring of Alzheimer's disease (works by G. Mazzoccoli, Kvetnoy, etc. ). This is important, since one of the urgent tasks of modern biomedicine is the development of methods (especially non-invasive) for life-time (and not post-mortem) diagnosis of Alzheimer's disease and other neurodegenerative pathologies.
After making these additions, I believe that the paper can be recommended for publication.
Reviewer 2 Report
Comments and Suggestions for Authors
The manuscript entitled “Biomarkers for managing neurodegenerative diseases” authored by Lara Cheslow et al aimed to review the current data regarding the availability and utility of biomarkers in neurodegenerative diseases management.
The amount of previously published data used for the review is generous (121 references), thus allowing the authors to produce a comprehensive description of the most prevalent neurodegenerative pathologies (Alzheimer’s disease, amyotrophic lateral sclerosis, and Parkinson’s disease), as well as of the available biomarkers and their use as prognostic and diagnostic indicators.
The manuscript introduction is very well constructed, with a clear presentation of the neurodegenerative diseases’ importance as the leading cause of cognitive and physical disability world-wide. Furthermore, the authors clearly state the importance of biomarkers as tools to detect the early stages of these diseases based on targeted pathogenesis.
For each selected pathology, the authors indicate using text and detailed scheme the diagnostic tool, sample type, the expected results of the biomarker level. Also, each section presenting the selected pathology presents biomarker use and misuse in therapies and clinical trials.
The conclusions are linked with objective prospects.
A very well written and scientific material !
My only suggestion is that the authors include the search protocol (databases, inclusion and exclusion criteria used to collect the presented data
